# Psychological well-being of SLE patients: A comprehensive examination of stress, depression, anxiety and quality of life

Pragya Singhal[1], Priyanka Srivastava[1]*, Liza Rajasekhar[2]

1 Perception and Cognition Lab, Cognitive Science Centre, Kohli Research Center on Intelligent Systems, International Institute of Information Technology, Hyderabad, Telangana, India, 2 Department of Clinical Immunology and Rheumatology, Nizam's Institute of Medical Sciences, Hyderabad, Telangana, India

* priyanka.srivastava@iiit.ac.in

## Abstract

Systemic Lupus Erythematosus (SLE/Lupus) is a chronic autoimmune disease affecting various organs, and associated with challenges in emotion and mood regulation. Despite the increasing prevalence of SLE globally and in India, neuropsychological data remain underrepresented. This study investigates the psychological health and quality of life among Indian SLE patients using self-reported questionnaires on depression, generalized anxiety, trait anxiety, stress, and quality of life. Results indicate a high prevalence of stress, depression, and both generalized and trait anxiety among the SLE patients. Additionally, SLE patients exhibited significantly reduced quality of life (physical well-being, social relationships, and environmental conditions) compared to their healthy counterparts. Disease severity was significantly correlated with patients' psychological health and quality of life, highlighting the substantial impact of the disease on individual well-being and their daily functioning. Further, cluster analysis revealed that older SLE patients with a homemaker background reported poorer psychological well-being compared with younger patients with a student background, highlighting the importance of demographic factors in understanding the SLE heterogeneity. In sum, poorer psychological well-being across demographic groups underscores the need for future research examining ethnic diversity within India and for translating these findings into holistic medical plans aimed at improving patients' quality of life.

## Introduction

Systemic Lupus Erythematosus (SLE) stands as a multifaceted autoimmune disease known for its diverse clinical manifestations, impacting mainly women and specific ethnic groups such as African or Asian populations [1]. Its dynamic nature yields a variable disease trajectory, necessitating continual monitoring to spot emerging or recurrent organ and systemic involvements. The complexity of SLE lies in its

**Data availability statement:** All data files are available from the Github database (urls: https://github.com/pragya919/Psychological-WellBeing-of-SLE-Patients).

**Funding:** The author(s) received no specific funding for this work.

**Competing interests:** The authors have declared that no competing interests exist.

heterogeneous presentation and the uncertain course it takes, making it a challenging disorder to diagnose and manage [2]. It is marked by a spectrum of shared symptoms like rashes across the cheeks and nose, anaemia, debilitating fatigue, hair loss, joint inflammation and swelling, susceptibility to blood clotting, and recurring headaches. The distinctiveness of SLE resides not only in its clinical diversity but also in its ability to impact other organs in the body, including the central nervous system (brain and spinal cord), adding complexity to diagnosis and treatment strategies.

The American College of Rheumatology (ACR) nomenclature system acknowledges the manifestation of neuropsychiatric disorders in SLE disease. Although the prevalence of reported neuropsychiatric syndromes of systemic lupus erythematosus (NPSLE) varies from 6% to 91% [3], the most commonly reported disorders are acute confused state, anxiety, mood and psychosis, and emotion and mood regulation [4–6]. In a recent study, 68% of SLE patients (n = 176) reported symptoms of depression, and 57.4% reported anxiety [7]. Notably, the reported psychological distress was observed in SLE patients even in the absence of apparent neuropsychiatric symptoms (non-NPSLE) [8]. When compared with patients with rheumatoid arthritis (another autoimmune disease), patients with SLE reported comparatively higher level of distress [8]. In sum, the high prevalence of depression and anxiety in SLE disease, regardless of overt neuropsychiatric symptoms, emphasizes the heterogeneous nature of the NPSLE and impedes the timely diagnosis. Further, the prevalence of psychological symptoms among healthy young adults [9,10] interferes with the attribution of psychological health challenges related to SLE.

Stress is another commonly reported neuropsychological health issue that intensifies the SLE disease and notably impacts the quality of life of those affected [11]. Stress was found to predict fatigue, depression, and pain among this group, indicating its substantial role in contributing to multiple symptoms [12]. The presence of depression and anxiety and their strong association with stress in SLE patients underscores the urgency for holistic management approaches to address psychological disorders to improve the well-being of individuals navigating this complex autoimmune condition.

Globally, the incidence of SLE ranges from 1.5 to 11 per 100,000 person-years, while the prevalence estimates range from 13 to 7,713.5 per 100,000 individuals [13], as reported in recent reviews representing populations from Asia, Australasia, North and South America, Europe, the UK, and Africa, [13–15]. These variations likely reflect differences in ethnicity, study settings, and environment. [13]. Despite this broad range of prevalence and incidence, there exists a significant dearth of comprehensive psychological data, particularly for the Asian and Indian populations.

Furthermore, the lack of linguistically agnostic neuropsychiatric screening tools, digitized health support, scarcity of neuropsychiatric professionals [16], societal stigma surrounding psychological health, socioeconomic constraints, and cultural recognition of such issues collectively compound the challenge and hinder effective monitoring and support for SLE patients in low and middle-income countries (LIMICs) like India. This notable absence raises critical concerns and emphasizes the urgent need for targeted research efforts to address the psychological facets of SLE among

the Indian population. Our study aims to assess the psychological well-being of Indian SLE patients using standardized assessments for various psychological domains.

## Methods

### Participants

G*Power version 3.1.9.6 [17] was used to calculate the minimum sample size required to test the study hypothesis. A F-test, one-way ANOVA (fixed effects, omnibus) was performed for three groups (namely cases, caregivers, and college students), using a medium effect size (0.25) and 80% power to achieve significance ($p < 0.05$). The sample size calculation indicated that a total of 159 participants ($n = 53$ for each participant group) would be required to examine the impact of SLE on psychological health and well-being scores. However, the sample size for SLE cases ($n = 102$) was deliberately increased to enable the use of advanced analyses, such as cluster analysis, to explore the heterogeneity within the SLE group.

In total, 207 participants were recruited for this study using a non-probability sampling method. Of these, 102 participants ($F = 92\%$, age $M = 26.84$ years, $SD = 5.98$ years), who fulfilled the 2012 SLE classification criteria [18] were enrolled as cases. The remaining 105 participants served as healthy controls. The healthy control group consisted of two subgroups: (a) the caregiver (CG) group, comprising 52 individuals ($F = 54\%$, age $M = 26.60$ years, $SD = 7.66$ years) who were caregivers of SLE patients and had no history of SLE; and (b) the college student (CS) group, which included 53 healthy college students ($F = 58\%$, age $M = 21.74$ years, $SD = 1.85$ years) with no personal history of autoimmune diseases.

The specific control groups were chosen to represent individuals from diverse cultural, linguistic, and socioeconomic backgrounds. The CG group was matched to the SLE group on key demographic characteristics, whereas the CS group was matched based on education level only, while differing in their socioeconomic, cultural, and linguistic backgrounds. These differences between the control subgroups allowed us to examine the potential role of demographic factors in psychiatric health reporting compared to SLE cases.

Inclusion criteria required participants to be adults aged between 18 and 45 years and to have completed at least higher secondary education. The upper age limit was chosen to minimize potential confounding effects of midlife factors and related life engagements on cognitive decline [19,20]. A minimum of higher secondary education was a pragmatic choice to ensure that participants could adequately read the instructions and consent form, and to reduce variability associated with education attainment, which is a widely discussed determinant of cognitive test performance [20,21]. The exclusion criteria for the control groups included any personal history of autoimmune diseases or other medical conditions that could affect psychological or cognitive health.

The current study is a part of ongoing research aimed at developing a language-agnostic neuropsychological test battery. The current data may serve as a valuable reference for future standardization efforts involving more diverse populations across varying age and education ranges.

Cases and the caregivers were recruited from the hospital, while college students were recruited from the university campus. Data collection began on 10 May 2023 and concluded on 7 March 2024. Participants did not receive any monetary compensation for their participation.

### Procedure

Each session began with an inclusion and exclusion criteria assessment, followed by the consent form. Detailed information about the study and the voluntary nature of participation, including the right to withdraw at any point without repercussions, was provided. Consent form sign-up was a two-step process. First, they were asked to 'read and sign' the printed consent form, which briefed them about the study, materials, the associated risks and benefits, the time taken to complete the session, and participants' right to withdraw at any point in time during the participation. Second, the participants were asked to click on the box to provide their consent to start the session. We included only data from those individuals (i.e.,

102 SLE cases and 105 healthy cohorts) who agreed and completed the study. We used the following statement to enable participants to provide their consent after asking for their name and DOB details.

Participants were asked to read the statement saying, "I confirm that I have read and understood the information sheet dated for the above study and have had the opportunity to ask questions. I understand that my participation in the study is voluntary and that I am free to withdraw at any time, without giving any reason, without my medical care or legal rights being affected. I hereby give my consent and give permission for members of the research team to have access to an anonymized version of the data and clinical information related to me to be used for research purposes only. I understand the anonymized data can be shared in the public domain with other researchers worldwide. I understand that my name will not be linked with the research materials, and I will not be identified or identifiable in the report or reports that result from the research. I agree to take part in the above study. I have read the above information and agreed to participate in this study."

The consent form, instructions, and questionnaires were available in both validated English and Hindi versions, based on the participants' language preference and convenience. These versions were chosen to ensure that participants fully understood the questions, thereby maintaining the accuracy and reliability of their responses.

Upon obtaining participant consent, the assessment commenced with a detailed demographic and health questionnaire. The psychological well-being questionnaires were randomly assigned to the participants to control the order effect. The experiment was performed in a quiet room, and the session lasted approximately 15–20 minutes. The study received approval from the respective institutional research ethics review board committees (IRB), i.e., Nizam's Institute of Medical Sciences (NIMS) (No.: EC/NIMS/3037/2022) and International Institute of Information Technology, Hyderabad (IIITH) IRB board (No.: IIITH-IRB-PRO-2023-08 dated 01-01-2023).

## Materials and Measures

The assessment comprised five distinct psychological questionnaires, including the Patient Health Questionnaire (PHQ-9) for depression, the Generalized Anxiety Disorder Assessment (GAD-7) for generalized anxiety, the Perceived Stress Scale (PSS-4) for stress, the Spielberger State-Trait Anxiety Inventory (STAI-T) for trait anxiety, and the World Health Organization Quality of Life Brief Assessment (WHOQOL-BREF), collectively aimed to offer a holistic evaluation of SLE related well-being issues.

### Patient Health Questionnaire – 9

The PHQ-9 questionnaire was designed as a concise, 9-item self-reported tool to evaluate an individual's experience of depression within the preceding two weeks, including the assessment date [22]. Each item of the questionnaire examined whether the respondent had been affected by specific problems or symptoms associated with depression during this time frame. Participants were instructed to carefully review each query and rate their experience on a 4-point scale: 0 indicating *not at all* and 3 signifying *nearly every day*. The scores for each response were tallied by summing the ratings across all nine items, resulting in a total score that spans a range from 0 to 27. The cut-off scores were 5, 10, 15 and 20 for mild, moderate, moderately severe and severe depression, respectively.

### Generalized anxiety disorder – 7

The GAD-7 questionnaire is an effective 7-item, self-reported tool designed to gauge an individual's generalized anxiety within the preceding two weeks, including the assessment date [23]. Like PHQ-9, GAD also used a similar frequency scale. The scale ranged from 0 = *not at all*, indicating the absence of the symptom, to 3 = *nearly every day*, denoting frequent occurrence. Upon completing the assessment, scores for each response were tabulated by summing the ratings across all seven items. This cumulative score represents a comprehensive evaluation of the severity of generalized anxiety experienced by the individual during the specified time frame, spanning a scoring range from 0 to 21. The cut-off scores were 5, 10 and 15 for mild, moderate and severe anxiety levels, respectively.

## Perceived stress scale – 4

In this study, we adopted the 4-item variant of PSS-10 due to time constraints and its user-friendly nature [24]. This abbreviated questionnaire aimed to assess the frequency of feelings experienced by participants over the preceding month. Participants were instructed to carefully consider each question and rate their responses on a 5-point scale, ranging from 0 = *never* to 4 = *very often*. For evaluation, the scores for the first and fourth questions are directly added, while responses for the second and third questions are reversed on the scale before summation. This method yields a total score ranging from 0 to 16, where higher scores signify elevated levels of perceived stress. A score greater than or equal to 6 was considered the cut-off for high stress.

## State-trait anxiety inventory – Trait

STAI is a widely used tool for assessing both state and trait anxiety. This study used a concise version consisting of 5 questions for each state and trait anxiety [25]. In our assessment, we only utilized the segment designed to measure trait anxiety. Trait anxiety refers to the tendency to experience anxiety as a personality characteristic [26]. Unlike generalized anxiety disorder, which is triggered by specific situations and involves distinct structural and functional patterns, trait anxiety is a more constant and pervasive aspect of an individual's personality [27]. Individuals with trait anxiety show more propensity to experience state anxiety [27]. Participants were instructed to devote sufficient time to contemplate each question and respond based on their general feelings. Participants expressed the extent to which each statement reflected their overall emotional state, using a 4-point scale ranging from 1 = *not at all* to 4 = *very much so*. Scores were computed by aggregating the responses across the 5 trait anxiety-related items, resulting in a total score ranging between 5 and 20. A score greater than or equal to 14 was considered the cut-off for high-trait anxiety.

## World Health Organization quality of life

The WHOQOL-BREF questionnaire, a shorter version of the original 100-item assessment, was chosen for our study to ensure efficiency and participant convenience. It comprised 26 items incorporating inquiries about the participant's feelings in the past two weeks, using varying response scales distributed across different domains [28]. Questions prompt individuals to evaluate 'how much', 'how completely', 'how often', 'how good', or 'how satisfied' they experienced aspects of their life [29]. With items evaluating physical health, psychological well-being, social relationships, and environmental aspects, this abbreviated version offers a more streamlined yet comprehensive assessment of an individual's quality of life.

## Statistical analysis

We used JASP version 0.95.3 [30] to perform the statistical analyses. The chi-squared test was used to analyse the demographics of the three participant groups. Furthermore, we conducted a multivariate analysis to assess the psychological test scores (GAD, PHQ, PSS and STAI-T) as dependent variables across the three participant groups: Cases, CG, and CS.

Box'M test yielded a $p = 0.724$, suggesting no violation of the multivariate homogeneity of variance. Additionally, multivariate normality was assessed using the Shapiro-Wilk test, adhering to the normalcy assumption ($p = 0.052$). Further, multicollinearity revealed moderate correlations among the psychological test scores, ranging from $\rho = 0.295$ to $\rho = 0.646$, meeting the assumption criteria. Adhering to the assumptions of homogeneity, multivariate normality, and multicollinearity, MANOVA was employed to analyze the psychological health scores, namely PHQ-9, GAD-7, STAI-T, and PSS-4, across the three participant groups [31].

However, a deviation from normalcy ($p < 0.05$) was observed for the individual psychological health scores. Consequently, we employed the Kruskal-Wallis test for the post-hoc analysis, followed by Dunn's test with Bonferroni correction

for the pair-wise comparisons. Similarly, the WHOQOL data violated the normalcy assumption and led us to choose the Kruskal-Wallis test to identify specific psychological domains that affected the QOL across three participant groups. Further, Dunn's post-hoc test with Bonferroni correction was used for pair-wise comparisons.

In addition to the comparative analysis, Spearman's correlational analysis explored the relationships between psychological health, quality of life, and disease severity. Moreover, we conducted a cluster analysis to identify the heterogeneity within our SLE patient population. We opted for the K-Means algorithm with four psychological test scores as parameters for defining our clusters. Mann-Whitney or chi-squared tests were used to compare the clusters based on the variable types. Finally, we determined the trends of psychological well-being among Indian SLE patients by calculating standard deviations (SD) for each patient's scores, using the mean and SD of both the health control groups as benchmarks for comparison.

## Results

The results section is structured into three main segments. The first segment focuses on the demographics of the participant groups. Secondly, the "Psychological Well-being Score" section compares the psychological health (depression, anxiety and stress) and quality of life scores among the SLE patients and the control groups. Lastly, the "SLE Disease and Psychological Well-being" section delves into the heterogeneity associated with the psychological well-being of SLE patients through correlational and cluster analysis. It explores the relationship between disease severity, psychological health, and quality of life in SLE patients. Additionally, cluster analysis was used to visualize the differences in the distribution of the patient population based on their psychological health scores. Finally, we determine the trends of psychological well-being in our patient population.

### Demographics

The analysis focused on categorical demographic variables, including gender, education, occupation, family income, and self-income (Table 1). Cases were demographically matched with the CG group based on education, culture, langauge, and family income. To capture additional diversity, including socioeconomic, cultural, linguistic, and university active life, as factors, we included a second control group comprising college-going young adults (i.e., CS group). The CS group differed from the SLE cases and CG in factors such as on-going college active life and family income, providing additional insights into the impact of these variables on psychological assessment. Notably, a significant proportion of the SLE cases comprised women working as homemakers. Consequently, differences emerged between the Cases and the healthy controls, particularly concerning gender, occupation, and self-income.

**Table 1. Demographics of participant groups: Cases (n = 102), CG (n = 52), CS (n = 53). Values are presented as percentage (rounded to whole numbers).**

| Demographics | Cases% | CG% | CS% |
|---|---|---|---|
| Gender, female | 92 | 54 | 58 |
| Highest Education | | | |
| Secondary | 19 | 10 | 0 |
| Higher Secondary | 37 | 38 | 72 |
| Tertiary | 39 | 46 | 24 |
| Post Graduation | 5 | 6 | 4 |
| Occupation | | | |
| Working | 16 | 39 | 0 |
| Non-working | 55 | 17 | 0 |
| Students | 29 | 44 | 100 |

Among the 102 cases, eighty-eight participants were outpatients and fourteen were inpatients at the time of assessment. The disease activity and damage were assessed using Systemic Lupus Erythematosus Disease Activity Index 2000 (SLEDAI-2K) and Systemic Lupus International Collaborating Clinics/ American College of Rheumatology Damage Index (SLICC/ACR-DI), respectively [32,33]. The SLEDAI-2K accounts for organ system involvement and symptoms over the preceding 30 days from the assessment date [32]. Higher SLEDAI-2K scores indicate increased disease activity. The SLEDAI-2K score ranges between 0 and 23 (*Mdn* = 3), and the SLICC/ACR damage index (SDI) ranges between 0 and 5 (*Mdn* = 0). Additionally, the BILAG scale was used to assess psychological issues associated with SLE cases [34]. Thirty-three SLE patients were identified as NP+ according to the BILAG scale [34] Clinical Characteristics of the patients are reported in Table 2.

## Psychological Well-being Score

### Psychological health score

Multivariate analysis (MANOVA) revealed a significant difference between the psychological health scores across participant groups *(F* (2,204) = 2.599; *p* = 0.009; *Wilk's* $\Lambda$ = 0.904). Kruskal-Wallis was performed separately on the psychological test scores for the three groups of participants. Significant differences were noted in GAD, PSS, and STAI-T scores (Table 3). Dunn's test with Bonferroni correction in the pair-wise analysis with the GAD and PSS revealed a significantly higher level of anxiety (*p* = 0.019) and stress (*p* = 0.019) in SLE patients compared to the caregivers (Fig 1). However, no significant difference was observed with college students (Fig 1). Trait anxiety (STAI-T scores) in SLE patients did not show significant differences with both control groups.

The raincloud plot (Fig 1) demonstrates the raw distribution of participants across the three groups and helps us visualize the distribution [35] of psychological health issues along with the mean difference between the three groups. The descriptive statistics show that around 40% of cases and college students reported moderate to severe anxiety (GAD score >= 10). However, only 30% caregivers reported a similar experience. Similarly, high trait anxiety (STAI-T score >= 14) was observed in 41% of cases but only 19% of college students and 17% of caregivers. In contrast, high stress

**Table 2. Clinical Characteristics of SLE Patients (n = 102). Values are presented as percentage (rounded to whole numbers).**

| SLE Categories | No. of Patients | Percentage% |
|---|---|---|
| NPSLE (BILAG) | 33 | 32 |
| Acute Cutaneous | 63 | 62 |
| Chronic Cutaneous | 21 | 20 |
| Oral or nasal ulcers | 52 | 51 |
| Non- scarring alopecia | 61 | 60 |
| Arthritis | 72 | 70 |
| Serositis | 24 | 23 |
| Renal | 48 | 47 |
| Neurologic | 23 | 22 |
| Hemolyticanemia | 13 | 13 |
| Leukopenia | 65 | 64 |
| Thrombocytopenia | 47 | 46 |
| ANA | 93 | 91 |
| Anti-dsDNA | 63 | 62 |
| Anti-Sm | 23 | 22 |
| Antiphospholoid antibodies | 12 | 11 |
| Low complement | 79 | 77 |

**Table 3. Psychological health and WHOQOL scores (*Mean* and *SD*), and Kruskal Wallis statistical values for the three groups, Cases, caregivers (CG), and college students (CS). Physical (Phy), Psychological (Psy), Social (Soc), and Environment (Env).**

| Variables | Cases (n = 102) | CG (n = 52) | CS (n = 53) | Kruskal Wallis H(df) | Rank $\epsilon^2$ |
|---|---|---|---|---|---|
| PHQ-9 | 8.15(5.77) | 6.78(5.02) | 8.50(5.06) | 3.22(2) | 0.02 |
| GAD-7 | 8.39(5.25) | 5.98(4.47) | 7.96(4.53) | 7.93(2)* | 0.02 |
| PSS-4 | 7.98(2.68) | 6.53(2.56) | 7.83(3.28) | 7.96(2)* | 0.04 |
| STAI-T | 11.84(4.13) | 10.25(3.66) | 10.40(3.49) | 7.21(2)* | 0.04 |
| WHOQOL Phy | 23.70(4.75) | 27.87(4.45) | 27.91(4.59) | 37.46(2)*** | 0.18 |
| WHOQOL Psy | 18.47(4.68) | 22.02(4.54) | 21.06(4.68) | 20.77(2)*** | 0.10 |
| WHOQOL Soc | 10.16(2.40) | 11.40(2.11) | 11.23(2.18) | 10.80(2)** | 0.05 |
| WHOQOL Env | 22.28(5.05) | 24.90(5.01) | 27.60(4.70) | 35.86(2)*** | 0.17 |

Note: * denotes $p < 0.05$; ** denotes $p < 0.01$; *** denotes $p < 0.001$

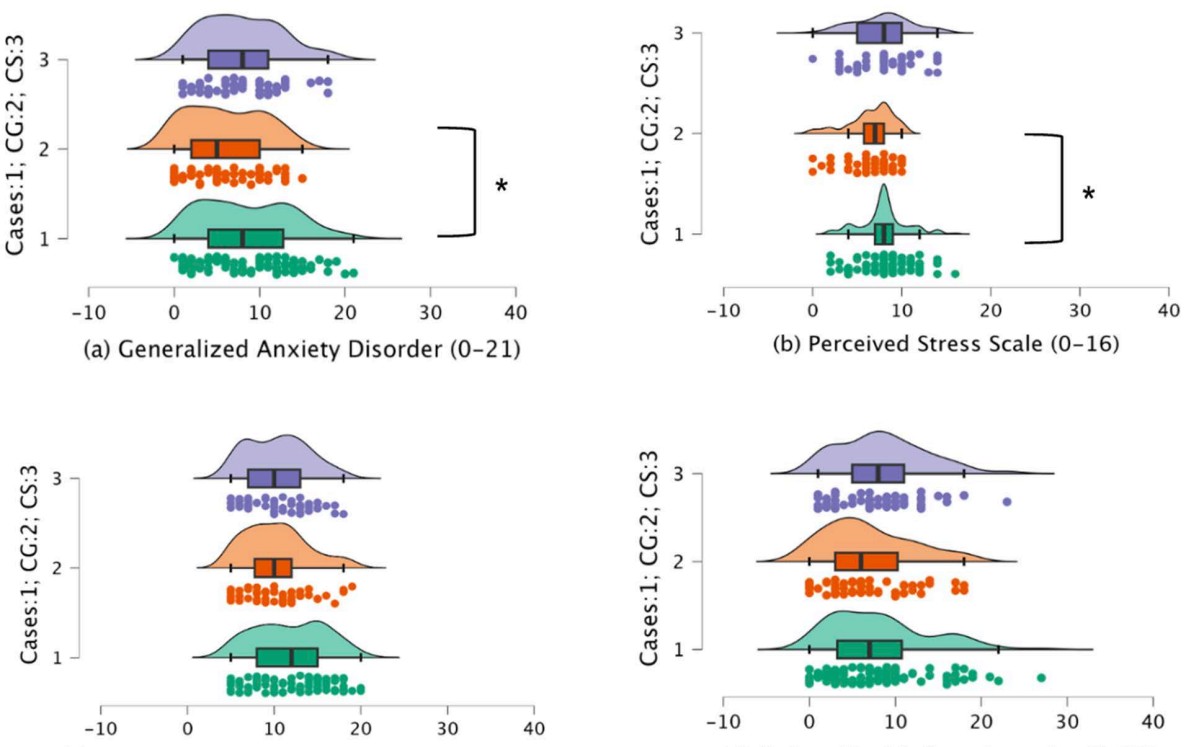

**Fig 1. Raincloud plot depicting scores for (a) Generalized Anxiety Disorder (GAD-7) (b) Perceived Stress Scale (PSS-4) (c) State-Trait Anxiety Inventory – Trait (STAI-T) (d) Patient Health Questionnaire (PHQ-9) for the three groups: Cases, caregivers (CG) and college students (CS).**
Note: * denotes $p < 0.05$; ** denotes $p < 0.01$; *** denotes $p < 0.001$.

(PSS score >= 6) was prevalent in more than 50% of all three participant groups, with 83% of cases, 75% of caregivers, and 73% of college students reported experiencing it in the last month. An almost equal level of depressive state was observed across the groups, with 35% of cases, 28% caregivers and 37% of college students reporting moderate to severe (PHQ score >= 10) depression symptoms.

### WHO quality of life score

The results revealed considerable differences in quality of life scores across all domains: physical, psychological, social, and environmental (Table 3). Post-hoc analysis showed significantly lower quality of life scores for SLE patients compared to control groups in the physical, psychological, social and environmental domains (Fig 2). Additionally, college students had significantly better environmental domain scores than caregivers.

### SLE disease and psychological well-being

#### Correlational analysis

Spearman's correlational analysis was conducted to examine the impact of disease severity on the psychological health and quality of life of SLE patients. Disease severity was quantified using SLEDAI-2K (mentioned in the Participants section under Methods). We observed a weak but significant positive correlation of SLEDAI-2K with GAD and PSS scores (Fig 3), suggesting a higher level of stress and anxiety in patients with increased disease severity. Further, the SLEDAI-2K score showed a weak but significant negative correlation with social and environmental domains of quality of life experiences by SLE patients, suggesting poorer social engagement and environmental conditions in patients with increased disease severity.

Further, we examined the relationship between the quality of life scores and the psychological well-being of the patients. There was a consistent negative correlation of all four domains of quality of life (physical, psychological, social and environmental) with the diverse psychological domains (GAD, PHQ, PSS and STAI-T) assessed in our study (Fig 3). A strong and significant negative correlation was observed for the physical domain of QOL with anxiety (GAD) and

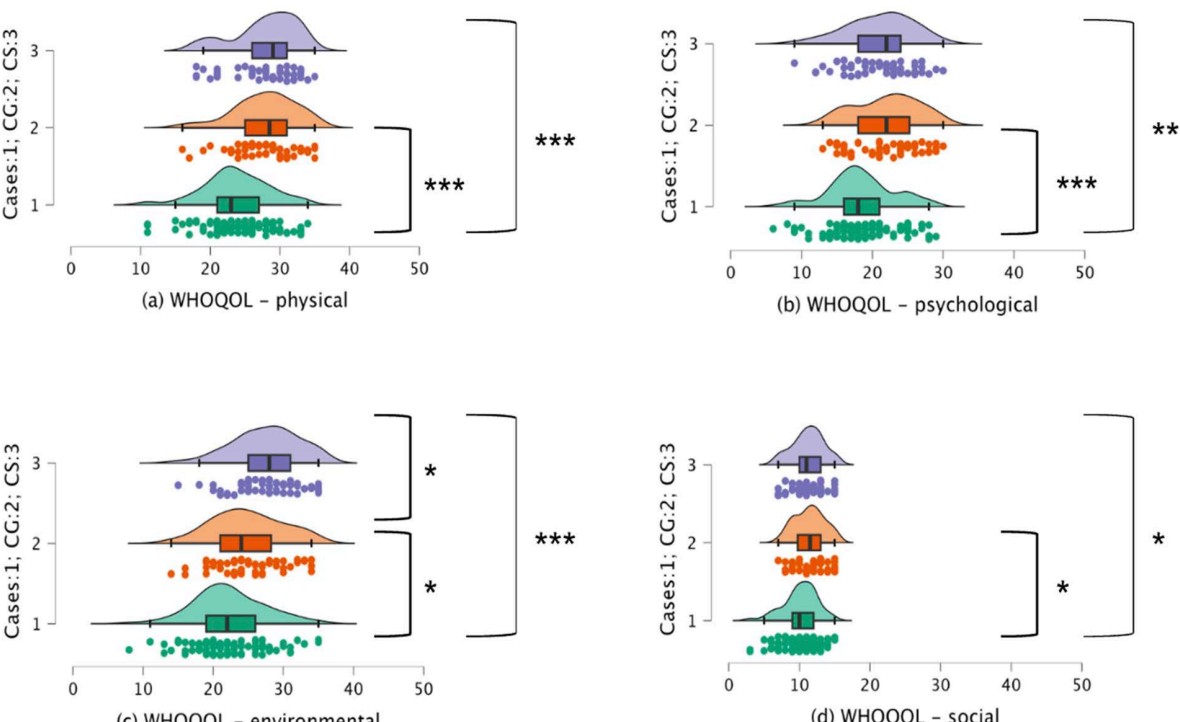

**Fig 2. Raincloud plot for Quality of Life Scores for (a) physical (b) psychological (c) environmental (d) social for the three groups: Cases, caregivers (CG) and college students (CS).** Note: * denotes $p < 0.05$; ** denotes $p < 0.01$; *** denotes $p < 0.001$.

**Fig 3. Spearman's coefficient (ρ) between psychological health, WHOQOL domains and SLEDAI-2K in SLE Patients.** Note: * denotes $p < 0.05$; ** denotes $p < 0.01$; *** denotes $p < 0.001$.

depression (PHQ). The psychological QOL domain also showed a strong correlation with depression (PHQ), trait anxiety (STAI-T), and generalized anxiety (GAD), indicating the construct validity of these measures.

## Cluster analysis

We utilized the K-Means algorithm to identify clusters based on four psychological scores: STAI-T, PHQ-9, GAD-7 and PSS-4. Training parameters included using the 'Means' Center type, the Hartigan-Wong algorithm, a maximum of 200 iterations and random sets, feature scaling, and setting the seed to 1. Two clusters were identified with a combined silhouette score of 0.380 (Cluster 1 = 0.410 and Cluster 2 = 0.338). Silhouette scores range from −1 to +1, with higher scores indicating better clustering quality [36]. Fig 4 shows the t-SNE plot for visualizing the cluster memberships [37]. Because of the violation of multivariate homogeneity ($p < 0.001$) and multivariate normality ($p = 0.005$) assumptions, we used the non-parametric Mann-Whitney test instead of multivariate analysis (MANOVA) to compare the two clusters. Results revealed that Cluster 2 exhibited poorer psychological health than Cluster 1, evidenced by their higher scores (Table 4) across all psychological tests (Fig 5).

Additionally, Cluster 2 showed lower WHOQOL scores in physical, psychological, social and environmental domains than Cluster 1 (Table 4). Demographically, Cluster 2 differed from Cluster 1 in terms of age ($p = 0.001$) and occupation ($p < 0.001$), with Cluster 2 mainly comprising older women who were predominantly unemployed (73% of Cluster 1 was unemployed). Clinically, Clusters 2 and 1 did not differ in the disease severity score (SLEDAI-2K), damage index (SDI), and neuropsychiatric involvement (NPSLE or non-NPSLE based on BILAG scores).

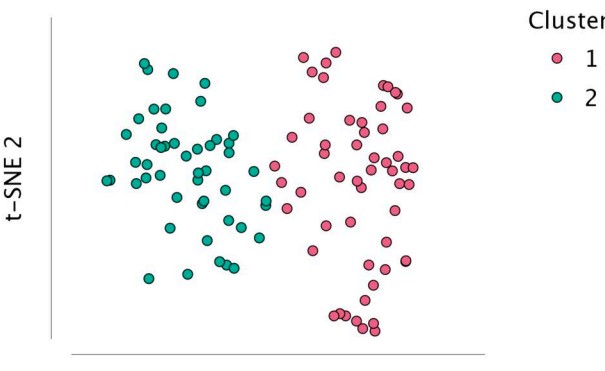

**Fig 4. t-SNE plot for clusters. Each coloured dot represents one sample in its respective cluster.**

**Table 4. Age, psychological health, and WHOQOL scores (*Mean and SD*) for clusters and effect size (Rank Biserial Correlation of Mann-Whitney test) for comparison between clusters. Physical (Phy), Psychological (Psy), Social (Soc), and Environment (Env).**

| Variables | Cluster 1 (n=54) | Cluster 2 (n=48) | Effect Size |
|---|---|---|---|
| Age | 25.29 (6.16) | 28.58 (5.33) | 0.37** |
| PHQ-9 | 4.65 (2.96) | 12.10 (5.62) | 0.77*** |
| GAD-7 | 4.24 (2.50) | 13.06 (3.18) | 0.97*** |
| PSS-4 | 7.07 (2.61) | 9.00 (2.41) | 0.41*** |
| STAI-T | 9.13 3.0s | 14.90 (2.87) | 0.81*** |
| WHOQOL Phy | 25.50 (4.37) | 21.67 (4.36) | −0.44*** |
| WHOQOL Psy | 20.35 (4.55) | 16.35 (3.91) | −0.51*** |
| WHOQOL Soc | 10.96 (2.21) | 9.25 (2.31) | −0.41*** |
| WHOQOL Env | 23.52 (5.12) | 20.90 (4.65) | −0.27* |

Note: * denotes $p < 0.05$; ** denotes $p < 0.01$; *** denotes $p < 0.001$

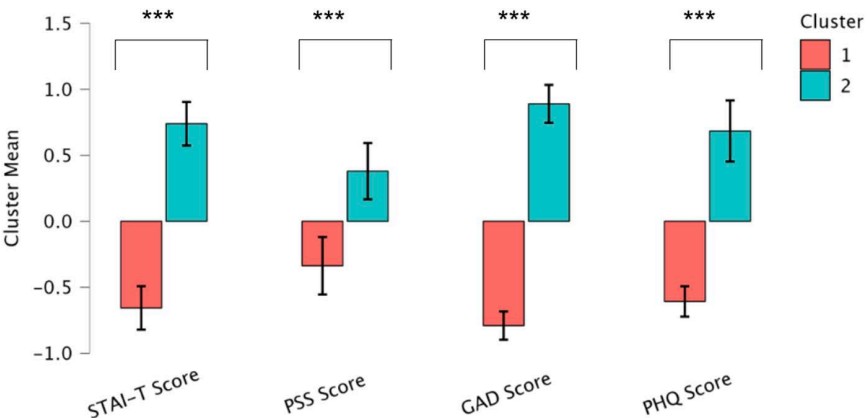

**Fig 5. Cluster Means for the psychological health scores of the two clusters.** Note: * denotes $p < 0.05$; ** denotes $p < 0.01$; *** denotes $p < 0.001$.

### Trends of psychological well-being in Indian SLE patients

We determined the standard deviation (SD) for each patient's psychological and quality of life scores by comparing them to the mean and standard deviation of two control groups: CG and CS. Our criterion for identifying psychological health deterioration was set at a threshold of greater than 1.5 standard deviations above the mean of the control groups for psychological health scores. Fig 6 illustrates the percentage of SLE patients exhibiting psychological decline in each domain based on this threshold. Compared to CG, 45% of SLE patients displayed symptoms for at least one of the four psychological disorders (depression, generalized anxiety, state anxiety and stress) evaluated in our study. In comparison, when compared to CS, 31% of SLE patients exhibited symptoms for at least one of these disorders.

### Discussion

The study examines the psychological well-being and quality of life experiences in SLE patients using caregivers and college students as controls. The results revealed elevated levels of stress and anxiety compounded with reduced quality of life (WHOQOL) experiences across all the domains (physical, psychological, social and environmental) in SLE cases compared to the control groups, CG and CS. Additionally, higher disease severity is associated with higher levels of stress and anxiety and poorer quality of social engagement and environment. This elaborated web of correlations suggests that factors contributing to mental health extend beyond psychological factors, incorporating aspects related to physical well-being, social interactions, and the environment. Further, results from cluster analysis highlight the heterogeneity associated with the psychological well-being of SLE patients, revealing distinct distributions within the patient population based on age and employment status. Despite observing a significant relationship between psychological well-being and SLE disease, the pervasive presence of psychological distress symptoms was evident across the groups (Fig 1 and Fig 2) in the current study. The results are discussed in light of previous findings and emerging trends in mental health worldwide and in India.

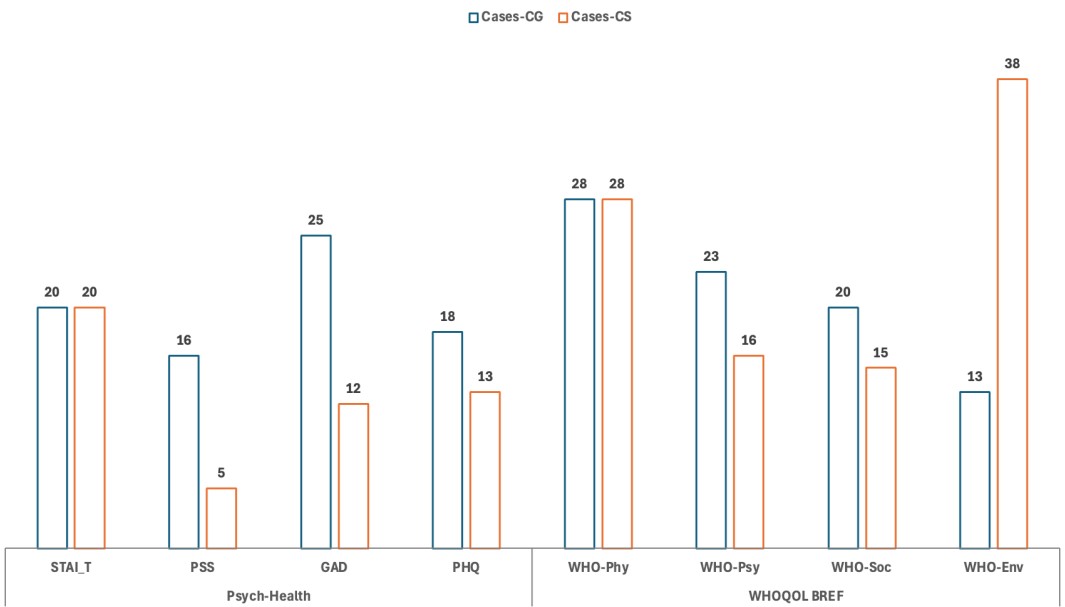

**Fig 6. Percentage of SLE patients with psychological health decline in each domain.** For standard psychological health scores, the criterion was >1.5 SD above the mean of the healthy controls. For the WHOQOL-BREF, it was defined as scores below the mean of healthy controls.

## Psychological Health and SLE

Our study highlights the high prevalence of self-reported psychological health issues in SLE patients, with more than 30% of the cases reporting experiences of depression, anxiety and stress as seen in previous studies [38,39]. Further, the association between disease severity and psychological health aligns with the previous findings and poses a serious concern regarding the cases' well-being and its association with their daily activities [39].

When compared to their healthy counterparts, SLE patients reported higher levels of anxiety and stress, aligning with previous results [38,40]. However, the current study fails to observe a significant difference between depressive symptoms across cases and healthy controls as seen previously [6]. One explanation could be the widespread nature of psychological health issues among young adults as seen in the raincloud plot. Recent studies [9,10] have raised concern regarding increasing trends of psychological distress among young adults. In India, it is suggested that nearly 15% of the population grapples with mental health issues at least once in their lifetime [9]. The pervasive prevalence and poor recognition of mental health issues among young adults not only weakens the recognition of SLE-induced psychological health concerns among patients but also impedes timely intervention.

## Psychological health and quality of life with SLE

Mental health is a complex, multifactorial issue. Among various factors, quality of life significantly correlates with an individual's mental well-being [41], particularly when someone is grappling with an autoimmune disease. SLE cases' poorer quality of life experiences across all domains suggest impaired everyday functioning. The compounding effect can be observed with disease severity, notably with the diminished quality of social engagement, an essential requirement for healthy living.

We also observed an association between psychological health and quality of life, suggesting that the better the quality of life is, the better the psychological health will be. By considering factors beyond psychological well-being, such as environmental, social, and physical influences, we gained a more holistic understanding of the overall well-being of SLE patients. The current practices in neuropsychiatric assessment do not focus on the quantitative and qualitative details of the psychological health issues and quality of life of the patients. The current findings recommend the inclusion of a comprehensive neuropsychiatric evaluation of SLE patients as part of regular health care practices at varying levels from primary to tertiary care.

## Heterogeneity in psychological Health of SLE patients

Results from the cluster analysis, although moderate (Silhouette score = *0.380*), suggest the presence of a distinct pattern in psychological health among the SLE cases. The moderate Silhouette score reflects the complex and interdependent nature of psychological symptoms in SLE cases. These variations were reflected in self-reported scores on depression, generalized anxiety, trait anxiety, and stress levels, with one cluster (i.e., Cluster 2) exhibiting elevated psychological symptoms than the other.

Cluster 2 (n = 48) comprised comparatively older participants (median age = 29 *vs.* 23 years) with a higher proportion of homemakers, resulting in fewer individuals with formal employment. Specifically, 73% of participants in Cluster 2 were homemakers, 17% were formally employed, and 10% were students, whereas Cluster 1 (n = 54) included 39% homemakers, 15% were formally employed participants, and 46% were students. The older age profile and higher proportion of participants without formal employment in Cluster 2 may help explain the poorer psychological health scores observed in this group compared with the younger, student dominated Cluster 1. These observations resonate with prior studies emphasizing the impact of unemployment on mental health, particularly in regions with limited economic development [42]. Additionally, participants in Cluster 2 reported lower quality of life across physical, psychological, social, and environmental domains, consistent with previous research linking quality of life to the mental health of SLE patients [43].

 

While these differences may not be directly clinically relevant, they provide insight into the demographic factors contributing to the quality-of-life experiences associated with SLE. These findings are consistent with previous reports highlighting demographic disparities in SLE burden, demonstrating higher incidence and prevalence rates of SLE among Black, Hispanic, and Asian populations compared with White Americans [13]. Overall, these findings highlight the heterogeneous nature of psychosocial well-being among SLE patients and emphasize the importance of considering demographic factors in future research aimed at elucidating the multifaceted contributors to this complexity.

Despite the promising results, a larger sample size would be required to enhance the robustness of the clusters, with multiple clustering methods used to better capture the underlying structure of complex psychological traits.

While psychological health scores on the PHQ-9, GAD-7, and WHO Quality of Life measures showed significant differences between the identified clusters, clinical factors such as disease severity (SLEDAI-2K) and the prevalence of neuropsychiatric manifestations (BILAG scores) did not reveal any significant differences across the identified clusters. This finding was unexpected, especially considering the known association between mental health disorders and central nervous system (CNS) involvement in SLE patients [43]. The lack of association between BILAG scores and psychological health scores may stem from the lack of detailed mental health and quality of life components within BILAG.

These results prompt a critical examination of existing medical protocols for NPSLE classification, which may not fully capture the complexity and nuances of neuropsychiatric conditions lacking factors such as temporal relationships of NPSLE to SLE diagnosis and disease severity as mentioned in previous studies [44]. Therefore, our study suggests the use of a multi-dimensional frequency-based psychological state assessment that enables a more comprehensive outlook of a patient than currently widely used medical protocols.

While current NPSLE medical protocols provide valuable information on the occurrence of neuropsychiatric incidents, they may fall short in offering insights into the frequency, quality, and severity of these occurrences. Therefore, there is a pressing need for an integrated examination that goes beyond binary categorizations and explores the intricate details of the patient experience. To further enrich our understanding, more investigations should delve into the heterogeneity associated with SLE and its nuanced impact on various psychological and quality-of-life domains.

### Psychological Health of Indian SLE patients

The trends revealed the presence of at least one psychological symptom in 45% of SLE patients compared to the caregivers. This underscores the substantial presence of psychological symptoms within the patient cohort when compared to the healthy control group matched for age, education, and socio-economic background. These findings are consistent with prior studies in India, which report similar trends in prevalence rates with depression and anxiety being the major symptoms [45,46]. Conversely, when compared against college students as the baseline, only 31% of the patients showed signs of at least one psychological symptom. This highlights the prevalence of mental health issues in young adults [9,47]. Hence, it is important to consider factors such as socioeconomic background and education when defining a healthy control group beyond merely the absence of the disease. Despite an increasing number of studies focusing on the Indian population, there remains a significant gap in well-defined normative baseline data. Future research endeavours are crucial to establish comprehensive baseline data to facilitate more accurate comparisons in this field.

### Limitations and Conclusion

Despite the use of cluster analysis to examine the heterogenous nature of psychological health issues within SLE population, we would like to note following caveats:

1. The current study is limited due to its narrow demographic profiles, particularly in terms education and age. Future studies should involve SLE cases across wide age range and educational backgrounds, as this would help characterize psychological symptoms and their variations among Indian SLE population.

2. Although the assumptions for MANOVA were met for multivariate analysis, the violation of normality in individual psychological health scores necessitated the use of non-parametric alternatives for post-hoc analyses. While multiple comparison corrections led the use of Bonferroni's corrections to control Type I error, it increases the risk of Type II errors, potentially reducing the chance to detect subtle effects. Future studies may address these concerns by increasing the sample size and/ or using a longitudinal research design.

3. Although K-means cluster analysis revealed a significant difference in psychological health and well-being between two clusters based on age and occupation, a larger sample size would be required to asses their clinical significance. Future studies could additionally employ multiple clustering methods along with sensitivity analysis to gain a more comprehensive understanding of the heterogeneity within SLE cases regarding psychological health.

In sum, our study is one of the first studies to provide a comprehensive assessment of the psychological health and quality of life among Indian SLE patients using validated instruments and two sets of healthy controls. Results reveal the presence of anxiety and stress among the SLE patient cohort when compared to their healthy counterparts. We explored the associations between psychological health, quality of life, disease factors and demographic factors. However, given the heterogeneous nature of SLE, a more nuanced analysis of the widespread psychological health issues through approaches like network or regression analysis is needed. Moreover, it's crucial to acknowledge the diverse spectrum of neuropsychiatric symptoms in SLE, ranging from emotional disorders to significant cognitive impairments. Previous studies have highlighted cognitive deficiencies across various domains critical for daily functioning, emphasizing the importance of investigating SLE's impact on cognitive health alongside psychological well-being. Incorporating cognitive health assessments will provide a more comprehensive understanding of the intricate relationship between cognitive and psychological health in individuals with SLE.

## Author contributions

**Conceptualization:** Priyanka Srivastava, Liza Rajasekhar.

**Data curation:** Pragya Singhal.

**Formal analysis:** Pragya Singhal, Priyanka Srivastava.

**Investigation:** Pragya Singhal, Priyanka Srivastava.

**Methodology:** Priyanka Srivastava.

**Project administration:** Priyanka Srivastava, Liza Rajasekhar.

**Resources:** Priyanka Srivastava, Liza Rajasekhar.

**Software:** Pragya Singhal.

**Supervision:** Priyanka Srivastava, Liza Rajasekhar.

**Validation:** Pragya Singhal.

**Visualization:** Pragya Singhal, Priyanka Srivastava.

**Writing – original draft:** Pragya Singhal, Priyanka Srivastava, Liza Rajasekhar.

**Writing – review & editing:** Priyanka Srivastava, Liza Rajasekhar.

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
