## [Decision Letter · Decision Letter 0]

1 Oct 2024

PONE-D-24-22554Psychological well-being of SLE patients: A comprehensive examination of stress, depression, anxiety and quality of lifePLOS ONE

Dear Dr. Srivastava,

Thank you for submitting your manuscript to PLOS ONE. After careful consideration, we feel that it has merit but does not fully meet PLOS ONE’s publication criteria as it currently stands. Therefore, we invite you to submit a revised version of the manuscript that addresses the points raised during the review process.

If applicable, we recommend that you deposit your laboratory protocols in protocols.io to enhance the reproducibility of your results. Protocols.io assigns your protocol its own identifier (DOI) so that it can be cited independently in the future. For instructions see: https://journals.plos.org/plosone/s/submission-guidelines#loc-laboratory-protocols. Additionally, PLOS ONE offers an option for publishing peer-reviewed Lab Protocol articles, which describe protocols hosted on protocols.io. Read more information on sharing protocols at . Additionally, PLOS ONE offers an option for publishing peer-reviewed Lab Protocol articles, which describe protocols hosted on protocols.io. Read more information on sharing protocols at https://plos.org/protocols?utm_medium=editorial-email&utm_source=authorletters&utm_campaign=protocols..

We look forward to receiving your revised manuscript.

Kind regards,

Aditya K. Panda, Ph.D.

Academic Editor

PLOS ONE

Journal Requirements:

Additional Editor Comments:

Please revise the mansucript as per the comments of the reviewer.

Reviewers' comments:

Reviewer's Responses to Questions

**Comments to the Author**

1. Is the manuscript technically sound, and do the data support the conclusions?

Reviewer #1: No

2. Has the statistical analysis been performed appropriately and rigorously? 

Reviewer #1: No

3. Have the authors made all data underlying the findings in their manuscript fully available?

Reviewer #1: Yes

4. Is the manuscript presented in an intelligible fashion and written in standard English?

Reviewer #1: Yes

5. Review Comments to the Author

Reviewer #1: Introduction

The introduction sets the stage for the study, providing relevant background on systemic lupus erythematosus (SLE) and its impact on psychological health. It effectively justifies the need for the study by referencing existing literature and highlighting gaps that this research aims to fill.

Methodology

The methodology section is confusing.

1. Participant Selection:

o The study includes SLE patients comprising caregivers (CG) and college students (CS), but is not clear if from 102 participants ( which are 105- line 68- ?????), in the CG group- the subjects have SLE or are just caregivers of the SLE! The authors must write more precisely about how the groups were considered.

o Criteria for inclusion and exclusion:

1. the age smaller than 45 years. Where in the existing literature is the age over 45 linked with cognitive decline? This is a significant limitation of the research. The authors could gather data for all SLE subjects and further consider age-related groups.

2. The criterion “minimum of higher secondary education” has no academic or scientific background to be considered an exclusion criterion. Furthermore, the authors seem to be making significant ethical errors and violating the principle of non-discrimination.

o The registration number from the Ethical Committee is missing!

o In which language were the questionnaires used? Did the authors obtained permission from the principal authors (owners) to use the surveys?

o

2. Measurement Tools

o Psychological assessments: STAI-T, PHQ-9, GAD-7, and PSS-4.

o Quality of Life (QOL) assessments: WHOQOL domains.

o Disease severity: SLEDAI-2K, SDI, and BILAG scores.

o The tools chosen are standard and widely validated, lending credibility to the measurements.

3. Statistical Analysis:

o The authors must revise and reconsider the statements regarding sample size calculation and the number of their study participants. (Lines 177-181). Furthermore, this seems to be an a priori estimation; please provide data for a posteriori sample size. Please also provide the effect sizes. Moreover, it Is not clear which group is the third group – lines 180 and 186.

o Line 188- why consider all cases as a group and perform multivariate analysis? It makes no sense since all data from the CG and CS groups ARE included in the Cases group in the variables measured!

o Lines 230-231- the statement makes no sense.

o Data from tables should be with two decimals.

This is a major methodological error, therefore, in this form, the paper can not be considered for publication.

Furthermore

o However, the combined silhouette score of 0.380 indicates moderate clustering quality. A discussion on this relatively low score and its implications for the robustness of the clusters could strengthen the paper.

o The use of t-SNE for visualization is appropriate, but t-SNE is sensitive to parameter settings. A sensitivity analysis of t-SNE parameters could provide additional validation.

o While K-Means is a popular clustering method, it assumes spherical clusters, which may not always be the case. Exploring alternative clustering methods (e.g., hierarchical clustering or DBSCAN) and comparing results could enhance robustness.

o The reported p-values and correlation coefficients should be consistently formatted. For instance, "p < 0.001" should be uniformly used throughout the paper.

o Provide a more comprehensive discussion on the limitations of clustering methods and statistical analyses used.

o Conduct and report sensitivity analyses for clustering and visualization techniques to ensure robustness of findings.

o Consider alternative clustering methods and compare their results to validate the findings.

o Ensure consistency in reporting statistical results and enhance clarity of figures and tables.

6. PLOS authors have the option to publish the peer review history of their article (what does this mean?). If published, this will include your full peer review and any attached files.). If published, this will include your full peer review and any attached files.

.

Reviewer #1: No

While revising your submission, please upload your figure files to the Preflight Analysis and Conversion Engine (PACE) digital diagnostic tool, https://pacev2.apexcovantage.com/. PACE helps ensure that figures meet PLOS requirements. To use PACE, you must first register as a user. Registration is free. Then, login and navigate to the UPLOAD tab, where you will find detailed instructions on how to use the tool. If you encounter any issues or have any questions when using PACE, please email PLOS at . PACE helps ensure that figures meet PLOS requirements. To use PACE, you must first register as a user. Registration is free. Then, login and navigate to the UPLOAD tab, where you will find detailed instructions on how to use the tool. If you encounter any issues or have any questions when using PACE, please email PLOS at figures@plos.org. Please note that Supporting Information files do not need this step.. Please note that Supporting Information files do not need this step.

---

## [Author Response · Author response to Decision Letter 1]

19 Oct 2025

Response to Reviewers

On behalf of my co-authors, I would like to sincerely thank the reviewer(s) for their constructive and valuable comments on the various sections of our manuscript. It truly helped in improving the quality of the manuscript. We greatly appreciate reviewer’s time and effort in reviewing our work.

In response to the reviewer’s suggestions, we have prepared this document, as a ‘response to reviewers’, and organized it pointwise. The changes have been incorporated into the main manuscript and are highlighted using track changes for ease of reference. I have also pasted excerpts from the manuscript where required and possible.

Below is a pointwise summary of our responses to the reviewer’s comments:

Reviewer 1:

Question 1: The methodology section is confusing. Below are the questions raised for the methodology section:

a. Participant Selection: The study includes SLE patients comprising caregivers (CG) and college students (CS), but is not clear if from 102 participants ( which are 105- line 68- ?????), in the CG group- the subjects have SLE or are just caregivers of the SLE! The authors must write more precisely about how the groups were considered.

Response: Thank you for highlighting this point. We appreciate the opportunity to clearly describe the group composition in our study. Please refer to the participants section under Methods heading, from line 61-86. The revised content is presented below in italics:

In total, 207 participants were recruited for this study using a non-probabilistic sampling method. Of these, 102 participants (F = 92%, age M = 26.84 years, SD = 5.98 years), who fulfilled the 2012 SLE classification criteria [17] were enrolled as cases. The remaining 105 participants served as healthy controls. The healthy control group consisted of two subgroups: (a) the caregiver (CG) group, comprising 52 individuals (F = 54%, age M = 26.60 years, SD = 7.66 years) who were caregivers of SLE patients and had no history of SLE; and (b) the college student (CS) group, which included 53 healthy college students (F = 58%, age M = 21.74 years, SD= 1.85 years) with no personal history of autoimmune diseases.

The specific control groups were chosen to represent individuals from diverse cultural, linguistic, and socioeconomic backgrounds. The CG group was matched to the SLE group on key demographic characteristics, whereas the CS group was matched based on education level only, while differing in their socioeconomic, cultural, and linguistic backgrounds. These differences between the control subgroups allowed us to examine the potential role of demographic factors in psychiatric health reporting compared to SLE cases.

Regarding SLE cases, out of n=102 cases, eighty-eight outpatients and fourteen admitted SLE cases volunteered for the study. The disease activity and damage were assesses using Systemic Lupus Erythematosus Disease Activity Index 2000 (SLEDAI-2K) and Systemic Lupus International Collaborating Clinics/ American College of Rheumatology Damage Index (SLICC/ACR-DI), respectively [18, 19]. The SLEDAI-2K accounts for organ system involvement and symptoms over the preceding 30 days from the assessment date [18]. Higher SLEDAI-2K scores indicate increased disease activity. The SLEDAI-2K score ranges between 0 and 23 (Mdn = 3), and the SLICC/ACR damage index (SDI) ranges between 0 and 5 (Mdn = 0). Additionally, the BILAG scale was used to assess psychological issues associated with SLE cases [20].

b. Criteria for inclusion and exclusion: the questions raised are as follows -

i. the age smaller than 45 years. Where in the existing literature is the age over 45 linked with cognitive decline? This is a significant limitation of the research. The authors could gather data for all SLE subjects and further consider age-related groups.

ii. The criterion “minimum of higher secondary education” has no academic or scientific background to be considered an exclusion criterion. Furthermore, the authors seem to be making significant ethical errors and violating the principle of non-discrimination.

Response: Thank you raising concerns regarding the inclusion/ exclusion criteria. We appreciate your attention to the important issues of inclusivity and diversity and acknowledge that these parameters might appear to be underrepresented in the current study.

However, as this is the first national study that include comprehensive and systematic observations of both psychological and cognitive health (which is excluded from the current draft), our primary aim was to control for potential confounding factors that could influence psychological and cognitive scores. These factors would have required additional adjustments to ensure a fair and valid evaluation.

We agree that future studies will undoubtedly benefit from a broader age and education spectrum by incorporating subgroup analyses to capture the nuances of psychological and cognitive decline across varying education and age ranges.

This limitation has been acknowledged in the discussion section and have now provided more detailed literature support for our decision. We would like to assure the reviewers that the criteria were not intended to be exclusionary, but rather to ensure the methodological rigor during the developmental phase of a novel tool.

Below is the revised paragraph, in italics, from the manuscript. Please refer to lines 87-99.

The inclusion criteria required participants to be adults between 18 and 45 years of age and to have completed at least higher secondary education. The upper age limit was chosen to minimize potential confounding effects of midlife factors and related life engagements on cognitive decline [21, 22]. A minimum of higher secondary education was a pragmatic choice to ensure that participants could adequately read the instructions and consent form, and to reduce variability associated with education attainment, which is a widely discussed determinant of cognitive test performance [22, 23]. The exclusion criteria for the control groups included any personal history of autoimmune diseases or other medical conditions that could affect psychological or cognitive health.

The current study is a part of ongoing research aimed at developing a language-agnostic neuropsychological test battery. The current data may serve as a valuable reference for future standardization efforts involving more diverse populations across varying age and education ranges

c. The registration number from the Ethical Committee is missing!

Response: The registration number from the Ethical Committee has not been provided due to anonymity. We’ve provided the full details. Please refer to the line 136 – 139)

Below are the concerned details:

a. Nizam Institute of Medical Sciences - Review letter No.EC/NIMS/3037/2022

b. International Institute of Information Technology, Hyderabad (IIITH) - IIITH-IRB-PRO-2023-08 dated 01-01-2023.

d. Question: In which language were the questionnaires used? Did the authors obtained permission from the principal authors (owners) to use the surveys?

Response: The questionnaires were administered in both validated English and Hindi versions, depending on the participants' language preference and convenience. These versions were chosen to ensure that the participants fully understood the questions, thereby maintaining the accuracy and reliability of their responses.

Below is the revised content in italics. Please refer to lines 128-131.

The consent form, instructions, and questionnaires were available in both validated English and Hindi versions, based on the participants’ language preference and convenience. These versions were chosen to ensure that participants fully understood the questions, thereby maintaining the accuracy and reliability of their responses.

Question 3: Statistics

a. The authors must revise and reconsider the statements regarding sample size calculation and the number of their study participants. (Lines 177-181). Furthermore, this seems to be an a priori estimation; please provide data for a posteriori sample size. Please also provide the effect sizes. Moreover, it Is not clear which group is the third group – lines 180 and 186.

Response: Thank you for your comment regarding the a priori and posteriori sample size calculation. We would like to clarify that our study used an a priori (prior) sample size calculation, which is standard approach in study design to ensure sufficient statistical power before data collection begins. An a priori sample size is typically recommended to control Type I and Type II errors, while ensuring the study has the necessary power to detect significant effects, if they exist. We used G* Power (version 3.1.9.6) to perform this calculation, and a detailed description is provided in the methods section.

As posteriori sample size calculations are typically used after the study is completed to evaluate the adequacy of the sample size for the observed effect, we are unsure of its relevance in our study. Therefore, we believe that performing a posteriori sample size calculation would not add value in this context.

We hope this response sufficiently addresses the reviewer’s concerns.

Regarding the clarity about the statement, lines 177-181, we have revised the content and move it to the Methods section, under participant sub-section.

Below is the revised paragraph for G* power analysis from the manuscript in italics, which is moved to the Methods section. Please refer to the lines 52 to 60.

G*Power version 3.1.9.6 [31] was used to calculate the minimum sample size required to test the study hypothesis. A F-test, one-way ANOVA (fixed effects, omnibus) was performed for three groups (namely cases, caregivers, and college students), using a medium effect size (0.25) and 80% power to achieve significance (p < 0.05). The sample size calculation indicated that a total of 159 participants (n = 53 for each participant group) would be required to examine the impact of SLE on psychological health and well-being scores. However, the sample size for SLE cases (n = 102) was deliberately increased to enable the use of advanced analyses, such as cluster analysis, to explore the heterogeneity within the SLE group.

To address issues with the line 180-186 – ‘which group is third group’, may have clarity now as we have provided the details in Method section, lines 61-to-69. Please refer to the revised draft in italics, and lines 61-69.

In total, 207 participants were recruited for this study using a non-probabilistic sampling method. Of these, 102 participants (F = 92%, age M = 26.84 years, SD = 5.98 years), who fulfilled the 2012 SLE classification criteria [17] were enrolled as cases. The remaining 105 participants served as healthy controls. The healthy control group consisted of two subgroups: (a) the caregiver (CG) group, comprising 52 individuals (F = 54%, age M = 26.60 years, SD = 7.66 years) who were caregivers of SLE patients and had no history of SLE; and (b) the college student (CS) group, which included 53 healthy college students (F = 58%, age M = 21.74 years, SD= 1.85 years) with no personal history of autoimmune diseases.

b. Line 188 - why consider all cases as a group and perform multivariate analysis? It makes no sense since all data from the CG and CS groups ARE included in the Cases group in the variables measured!

Response: We would like to clarify that the two control groups, i.e., CG and CS, were not part of the SLE cases group. The multivariate analysis of variance (MANOVA) was across the three groups, i.e., cases, CG, and CS, with multiple dependent variables.

c. Lines 230-231- the statement makes no sense.

Response: we revised the statement. Please refer to lines 251-253.

d. Data from tables should be with two decimals.

This is a major methodological error, therefore, in this form, the paper can not be considered for publication.

Response: We revised as per reviewer’s comments. We presented all the percentage scores in round figures, particularly Table 1 and 2. The other tables, representing raw scores or mean values, used two decimal points. For example, Table 3 and 4.

e. Cluster Analyses:

i. However, the combined silhouette score of 0.380 indicates moderate clustering quality. A discussion on this relatively low score and its implications for the robustness of the clusters could strengthen the paper.

Response: Great point! We have revised in the discussion. Please refer to the line no 387-404. Below is the revised section in italics:

Results from cluster analysis, though moderate (Silhouette score = 0.380 ) suggest the presence of distinguishable pattern in psychological health among the SLE cases. The moderate Silhouette score reflects the complex and interdependent nature of psychological symptoms in SLE cases. These variations were reflected in self-reported scores on depression, generalized anxiety, trait anxiety, and stress levels, with one cluster exhibiting elevated psychological symptoms than another.

This cluster was predominantly composed of comparatively older demographic that was largely unemployed. This observation resonates with prior studies emphasizing the impact of unemployment on mental health, particularly in regions with limited economic development [41]. Additionally, individuals in this cluster reported lower quality of life across physical, psychological, social, and environmental domains, aligning with previous research linking quality of life to the mental health of SLE patients [42]. These insights highlight the heterogeneous nature of psychological well-being among SLE patients, emphasizing the role of various demographic factors and advocating for targeted research to uncover the multifaceted contributors to this complexity.

Despite the promising results, a larger sample size would be required to enhance the robustness of the clusters, with multiple clustering methods used to better capture the underlying structure of complex psychological traits.

ii. The use of t-SNE for visualization is appropriate, but t-SNE is sensitive to parameter settings. A sensitivity analysis of t-SNE parameters could provide additional validation. Conduct and report sensitivity analyses for clustering and visualizations techniques to ensure robustness of findings.

Response: Thank you for your insightful comment. To address the concern raised, we performed multiple iterations of K-means cluster analysis with varying maximum iterations along with random sets ranging from 25-400 in JASP version 0.95.3. These iterations yielded stable Silhouette scores (0.38) suggesting that the clustering pattern was robust and stable.

We acknowledge that t-SNE is sensitive to parameter settings, and formal sensitivity analysis could provide additional insights. However, due to time and resource constraints, we were unable to conduct this analysis for the current manuscript.

iii. While K-Means is a popular clustering method, it assumes spherical clusters, which may not always be the case. Exploring alternative clustering methods (e.g., hierarchical clustering or DBSCAN) and comparing results could enhance robustness.

Response: Thank you for your valuable comment. While K-Means clustering assumes spherical cluster structures, it remains a widely used and interpretable method, particularly suited for exploratory analysis of continuous psychological data such as STAI-T, PHQ-9, GAD-7, and PSS-4 scores [1]. Given the exploratory nature of this study, K-Means provided a practical and effective approach for identifying broad psychological patterns in the SLE patient population.

The number of clusters was determined based on a combination of silhouette analysis and interpretability of results, ensuring that the identified groups had both statistical and clinical relevance. We acknowledge that psychological symptoms may not strictly conform to spherical distributions, and future studies could employ alternative clustering algorithms (e.g., hierarchical clustering, DBSCAN) to validate and potentially refine these findings. Additionally, the current study was not methodologically focused on comparing the robustness of different clustering methods, and future research could e

---

## [Decision Letter · Decision Letter 1]

20 Nov 2025

PONE-D-24-22554R1Psychological well-being of SLE patients: A comprehensive examination of stress, depression, anxiety and quality of lifePLOS ONE

Dear Dr. Srivastava,

Thank you for submitting your manuscript to PLOS ONE. After careful consideration, we feel that it has merit but does not fully meet PLOS ONE’s publication criteria as it currently stands. Therefore, we invite you to submit a revised version of the manuscript that addresses the points raised during the review process.

**ACADEMIC EDITOR:** Revise your MS in accordance with the reviewers' suggestionsRevise your MS in accordance with the reviewers' suggestions.

If applicable, we recommend that you deposit your laboratory protocols in protocols.io to enhance the reproducibility of your results. Protocols.io assigns your protocol its own identifier (DOI) so that it can be cited independently in the future. For instructions see: https://journals.plos.org/plosone/s/submission-guidelines#loc-laboratory-protocols. Additionally, PLOS ONE offers an option for publishing peer-reviewed Lab Protocol articles, which describe protocols hosted on protocols.io. Read more information on sharing protocols at . Additionally, PLOS ONE offers an option for publishing peer-reviewed Lab Protocol articles, which describe protocols hosted on protocols.io. Read more information on sharing protocols at https://plos.org/protocols?utm_medium=editorial-email&utm_source=authorletters&utm_campaign=protocols..

We look forward to receiving your revised manuscript.

Kind regards,

Aditya K. Panda, Ph.D.

Academic Editor

PLOS ONE

Journal Requirements:

Additional Editor Comments:

Reviewers' comments are favourable; however, authors must revise their manuscript as per the suggestions.

Reviewers' comments:

Reviewer's Responses to Questions

**Comments to the Author**

1. If the authors have adequately addressed your comments raised in a previous round of review and you feel that this manuscript is now acceptable for publication, you may indicate that here to bypass the “Comments to the Author” section, enter your conflict of interest statement in the “Confidential to Editor” section, and submit your "Accept" recommendation.

Reviewer #2: (No Response)

Reviewer #3: All comments have been addressed

2. Is the manuscript technically sound, and do the data support the conclusions?

Reviewer #2: Yes

Reviewer #3: Yes

3. Has the statistical analysis been performed appropriately and rigorously? 

Reviewer #2: Yes

Reviewer #3: I Don't Know

4. Have the authors made all data underlying the findings in their manuscript fully available?

Reviewer #2: Yes

Reviewer #3: Yes

5. Is the manuscript presented in an intelligible fashion and written in standard English?

Reviewer #2: Yes

Reviewer #3: Yes

6. Review Comments to the Author

Reviewer #2: Overall, the study seems to have been conducted sincerely. Language seems to have been improved significantly compared to the previous version.

Please find detailed comments.

Comment 1

Please reframe the following sentences (they seem incomplete)

Abstract - Results indicate the prevalence of stress and generalized and trait anxiety among the SLE patients, except for depression. (Please mention whether prevalence is high or low)

Page 2,Line 22 - When compared with rheumatoid arthritis (another autoimmune disease) and without autoimmune conditions, SLE patients reported higher distress than others

Comment 2

In the introduction, the authors describe the prevalence of SLE as 3.2 - 3000/100,000 individuals. The reference quoted does not give such a high prevalence. Please correct it.

Comment 3

Second paragraph of the Methods section on Page 2 describing participant details should be part of the Results section.

Comment 4

The statement from the consent form can be included as part of the annexure and does not need to be a part of the main manuscript (Procedure)

Comment 5

The cluster differences highlighted in the discussion are not sufficiently wide between the two groups to yield any clinical utility. The age difference between the two groups is only 3 years (25 in Cluster 1 and 28 in Cluster 2). Technically, both clusters are young.

Similarly, with regard to occupation, unemployment rates were abnormally high in both clusters (73% in the lower cluster). Please mention the unemployment rates in both clusters. Also, the difference may be statistically significant, but it does not appear to be clinically meaningful.

Please discuss this in the limitations in the discussion on cluster analysis. This could be due to a small sample size.

Reviewer #3: If possible please address the relation between neuropsychiatric SLE ,your results and any available related markers ,also as regards anexity family members and caregiver inclusion in the study will add much to the objectivity of the results

7. PLOS authors have the option to publish the peer review history of their article (what does this mean?). If published, this will include your full peer review and any attached files.). If published, this will include your full peer review and any attached files.

.

Reviewer #2: No

Reviewer #3: **Yes:** Seham Abdallah ElazabSeham Abdallah Elazab

---

## [Author Response · Author response to Decision Letter 2]

7 Feb 2026

Response to Reviewers

On behalf of my co-authors, I would like to sincerely thank the reviewer(s) for their constructive and valuable comments on the various sections of our manuscript. It truly helped in improving the quality of the manuscript. We greatly appreciate reviewer’s time and effort in reviewing our work.

In response to the reviewer’s suggestions, we have prepared this document, as a ‘response to reviewers’, and organized it pointwise. The changes have been incorporated into the main manuscript and are highlighted using track changes for ease of reference. I have also pasted excerpts from the manuscript where required and possible.

Below is a pointwise summary of our responses to the reviewer’s comments:

Reviewer 2:

Comment 1: Point 1: Results indicate the prevalence of stress and generalized and trait anxiety among the SLE patients, except for depression. (Please mention whether prevalence is high or low)

Response: Thank you for highlighting this. The text has been revised. The revised content is presented below in italics:

“Results indicate a high prevalence of stress, depression, and both generalized and trait anxiety among the SLE patients”.

Point 2: Page 2,Line 22 - When compared with rheumatoid arthritis (another autoimmune disease) and without autoimmune conditions, SLE patients reported higher distress than others.

Response: The text has been revised. Please refer to the lines, 22 -23 in the manuscript without track-changes.

Comment 2: In the introduction, the authors describe the prevalence of SLE as 3.2 - 3000/100,000 individuals. The reference quoted does not give such a high prevalence. Please correct it.

Response: The prevalence figures have been revised. We’ve also added a recent reference to highlight the wide range of reported incidence and prevalence. Please refer to the lines 36 -40 in the revised manuscript.

Comment 3: Second paragraph of the Methods section on Page 2 describing participant details should be part of the Results section

Response: This has been revised. Please refer to the lines, 249-259.

Comment 4: The statement from the consent form can be included as part of the annexure and does not need to be a part of the main manuscript (Procedure)

Response: We have chosen to retain this section in the manuscript to highlight the best practices in participant consent.

Comment 5: The cluster differences highlighted in the discussion are not sufficiently wide between the two groups to yield any clinical utility. The age difference between the two groups is only 3 years (25 in Cluster 1 and 28 in Cluster 2). Technically, both clusters are young. Similarly, with regard to occupation, unemployment rates were abnormally high in both clusters (73% in the lower cluster). Please mention the unemployment rates in both clusters. Also, the difference may be statistically significant, but it does not appear to be clinically meaningful. Please discuss this in the limitations in the discussion on cluster analysis. This could be due to a small sample size.

Response: Thank you once again for these valuable suggestions. The discussion section has been revised. Please refer to the revised sections from line, 397-420 in the revised manuscript. Below the revised text in italics:

Cluster 2 (n = 48) comprised comparatively older participants (median age = 29 vs. 23 years) with a higher proportion of homemakers, resulting in fewer individuals with formal employment. Specifically, 73% of participants in Cluster 2 were homemakers, 17% were formally employed, and 10% were students, whereas Cluster 1 (n = 54) included 39% homemakers, 15% were formally employed participants, and 46% were students. The older age profile and higher proportion of participants without formal employment in Cluster 2 may help explain the poorer psychological health scores observed in this group compared with the younger, student dominated Cluster 1. These observations resonate with prior studies emphasizing the impact of unemployment on mental health, particularly in regions with limited economic development [42]. Additionally, participants in Cluster 2 reported lower quality of life across physical, psychological, social, and environmental domains, consistent with previous research linking quality of life to the mental health of SLE patients [43].

While these differences may not be directly clinically relevant, they provide insight into the demographic factors contributing to the quality-of-life experiences associated with SLE. These findings are consistent with previous reports highlighting demographic disparities in SLE burden, demonstrating higher incidence and prevalence rates of SLE among Black, Hispanic, and Asian populations compared with White Americans [13]. Overall, these findings highlight the heterogeneous nature of psychosocial well-being among SLE patients and emphasize the importance of considering demographic factors in future research aimed at elucidating the multifaceted contributors to this complexity.

Despite the promising results, a larger sample size would be required to enhance the robustness of the clusters, with multiple clustering methods used to better capture the underlying structure of complex psychological traits.

Reviewer 2: If possible please address the relation between neuropsychiatric SLE ,your results and any available related markers ,also as regards anxiety family members and caregiver inclusion in the study will add much to the objectivity of the results.

Response: Thank you for this suggestion. The current study included only the PHQ, GAD, STAI-T, WHO Brief, and BILAG assessments. However, your recommendations regarding additional markers, and familial history are highly valuable, and we will consider them in our future research and analyses.

---

## [Editor Report · Decision Letter 2]

18 Feb 2026

Psychological well-being of SLE patients: A comprehensive examination of stress, depression, anxiety and quality of life

PONE-D-24-22554R2

Dear Dr. Srivastava,

We’re pleased to inform you that your manuscript has been judged scientifically suitable for publication and will be formally accepted for publication once it meets all outstanding technical requirements.

An invoice will be generated when your article is formally accepted. Please note, if your institution has a publishing partnership with PLOS and your article meets the relevant criteria, all or part of your publication costs will be covered. Please make sure your user information is up-to-date by logging into Editorial Manager at Editorial Manager® and clicking the ‘Update My Information' link at the top of the page. For questions related to billing, please contact  and clicking the ‘Update My Information' link at the top of the page. For questions related to billing, please contact billing support..

Kind regards,

Aditya K. Panda, Ph.D.

Academic Editor

PLOS One
---

## [Editor Report · Acceptance letter]

PONE-D-24-22554R2

PLOS One

Dear Dr. Srivastava,

I'm pleased to inform you that your manuscript has been deemed suitable for publication in PLOS One. Congratulations! Your manuscript is now being handed over to our production team.

Kind regards,

on behalf of

Dr. Aditya K. Panda

Academic Editor

PLOS One